# Assessing the Hands-on Usability of the Healthy Jeart App Specifically Tailored to Young Users

**DOI:** 10.3390/healthcare12030408

**Published:** 2024-02-05

**Authors:** Ana Maria Roldán-Ruiz, María-de-los-Ángeles Merino-Godoy, Antonio Peregrín-Rubio, Carmen Yot-Dominguez, Emília Isabel Martins Teixeira da Costa

**Affiliations:** 1Information Technologies Department, School of Engineering, University of Huelva, 21007 Huelva, Spain; amroldan@dti.uhu.es; 2Nursing Department, Faculty of Nursing, University of Huelva, 21007 Huelva, Spain; 3Centre for Advanced Studies in Physics, Mathematics and Computing, Andalusian Inter-University Institute in Data Science and Computational Intelligence, University of Huelva, 21007 Huelva, Spain; peregrin@dti.uhu.es; 4Didactics and Educational Organization Department, Faculty of Education Sciences, University of Seville, 41001 Seville, Spain; carmenyot@us.es; 5Nursing Department, Health School, University of Algarve, 8000 Faro, Portugal; eicosta@ualg.pt; 6Health Sciences Research Unit: Nursing (UICISA: E), Nursing School of Coimbra (ESEnfC), 3000 Coimbra, Portugal

**Keywords:** usability, adolescents, m-health, health promotion

## Abstract

Background: The widespread adoption of mobile devices by adolescents underscores the potential to harness these tools to instill healthy habits into their daily lives. An exemplary manifestation of this initiative is the Healthy Jeart app, crafted with the explicit goal of fostering well-being. Methodology: This study, framed within an applied investigation, adopts an exploratory and descriptive approach, specifically delving into the realm of user experience analysis. The focus of this research is a preliminary examination aimed at understanding users’ perceived usability of the application. To glean insights, a comprehensive questionnaire was administered to 101 teenagers, seeking their evaluations on various usability attributes. The study took place during 2022. Results: The findings reveal a considerable consensus among users regarding the evaluated usability aspects. However, the areas for improvement predominantly revolve around managing the information density, particularly for a subset of end users grappling with overwhelming content. Additionally, recommendations are put forth to streamline the confirmation process for user suggestions and comments. Conclusion: This analysis illuminates both the strengths of the app and areas ripe for refinement, paving the way for a more user-centric and efficacious Healthy Jeart application.

## 1. Introduction

### 1.1. Healthy Behaviors among Young People

Adolescence is a key stage in the promotion of a healthy life [1]. The family’s influence is essential in promoting healthy behaviors, yet schools, serving as both a social environment for peer interaction and a formal learning setting, undeniably serve as pivotal scenery for acquiring healthy habits. These habits will significantly shape the future health of young individuals [2].

Several studies revealed the relationship between unhealthy lifestyle habits such as dietary imbalances and the onset of non-communicable diseases such as diabetes, cancer, or cardiovascular pathologies including high blood pressure and hypercholesterolemia, among others [3,4]. Recent data seem to indicate that in Spain, the young population frequently fails to follow a balanced diet and the recommended levels of physical activity [5,6,7,8]. Hence, it is essential to gradually devise interventions that prioritize health enhancement using technology, specifically mobile applications, serving as a highly effective platform for implementing these initiatives. In this regard, the World Health Organization [9] recently published a guide on their design, development, and implementation.

In this scenario, it appears crucial to delve into the relationship between technology and health, recognized as a promising combination for fostering healthy habits during these formative years [10]. Presently, there are a vast array of health-related informatic applications available [11,12]. Yet, there remains room for improvement in the overall quality of the available apps designed to enhance dietary choices and physical activity and reduce sedentary behavior among children and adolescents [13,14].

In this context, the “Healthy Jeart” project emerged. It is an application specifically created to be used by children and young people, both inside and outside of school. The aim is to foster the knowledge, attitudes, and healthy habits that should proliferate in a young population of 8–16-year-olds. To achieve this, it employs communication styles suitable for specific ages and incorporates enjoyable components [15]. Additionally, it delivers straightforward and easily comprehensible messages offering advice and tips across various health domains and fostering healthy habits (Figure 1 and Figure 2). Healthy Jeart is tailored to address the needs and interests of young individuals. Its creation was initiated by forming nominal groups, pinpointing the most pertinent subjects that resonated with them [16].

The Healthy Jeart app has been acknowledged by the regional state Agency of Health Quality of Andalusia with the “Healthy App” distinction [16] and is also endorsed by the Association of Community Nursing (AEC) in the Spanish national scope. Likewise, its associated website has received the “Advanced Level Healthcare Website” accreditation seal, granted by the Andalusian Regional Agency for Healthcare Quality [17]. Additional details can be accessed on the application’s website: https://www.healthyjeart.com (accessed on 7 January 2024).

### 1.2. Digital Applications and the Importance of Their Usability

Mobile applications, as a type of software, need to meet users’ explicit and implicit needs for acceptance. In software engineering, which covers the entire lifecycle of computer application development, the central emphasis is on creating high-quality products regardless of the device. This highlights the crucial role of design and production quality.

In software development, quality entails meeting essential characteristics, with usability being a pivotal feature. Usability examines how effectively users can achieve goals, considering factors like effectiveness, efficiency, and satisfaction. Evaluation includes aspects such as cognitive load, learnability, and software portability across platforms [18,19,20].

Evaluating usability encompasses testing how easily and effectively apps can be used in both real-world settings and controlled lab environments, and these conditions significantly impact the outcomes [21]. Some authors opt to conduct usability assessments consistently across the entire software product’s lifecycle. They evaluate it using reports at each stage of development rather than solely at the conclusion of the development phase [22]. Specialists analyzing final products [23,24] conduct tests in controlled environments using questionnaires with closed or open-ended questions. A scoring scale is used to assess user perceptions. Notably, there is a growing focus on researching the usability of applications designed for educational settings within this domain [25].

It is along these lines that Zhou and col. [26] worked to measure the usability of m-health applications. While there exist various methods for conducting usability studies [27], it is crucial to highlight the work of Kumar, Goundar, and Chand [28]. Their contribution offers a contemporary, design-focused framework for assessing and studying usability in mobile learning applications, encompassing elements spanning from content structuring to navigation. This source contributed as a cornerstone in our theoretical framework.

Briefly put, the creation of educational apps necessitates a usability analysis to ensure their alignment with both pedagogical and technological standards, merging criteria outlined by software engineering and pedagogy [29].

Usability is thus an essential element that ensures that users can make proper use of applications to achieve their objectives without problems. Hence, this project focuses on refining the usability of the Healthy Jeart application. Its primary aim is to pinpoint any usability concerns, grasp the user requirements, and ultimately enhance the overall user-friendliness of the product. Evaluation of the usability of Healthy Jeart was carried out empirically, with real users (school-age adolescents).

## 2. Materials and Methods

### 2.1. Study Design, Setting, and Participants

This is an exploratory and descriptive study that is part of an applied investigation, specifically within the domain of user experience analysis. The population consisted of 190 primary and secondary school students from a public–private school in the province of Andalusia (Spain), who installed the Healthy Jeart app on their electronic devices. Convenience sampling was selected based on the practical considerations arising from constraints on time and resources. Given the nature of our study, which necessitated the involvement of individuals with specific experiences—such as using the app over a defined period—convenience sampling provided a straightforward method to access participants meeting these criteria, as they were readily available within our immediate surroundings.

Based on the information provided by the Regional Ministry of Education and Sport for the academic year 2019–2020 [30], Andalusia had a total of 1,500,265 students enrolled in non-university general education. Among them, 21.9% (328,636) attended subsidized centers, which is the focus of this study. Additionally, during this period, the region had 658,084 young individuals aged between 11 and 17, with 51.3% of them being boys.

Prior to the commencement of the study, the necessary official approval was diligently acquired. The ethical considerations and protocols were formally reviewed and approved by the Research Ethics Committee of the Province of Huelva, under the protocol code PI047/16. This ethical clearance ensured that the study adhered to the highest standards of ethical conduct in research. Following the receipt of ethical approval, an extensive demonstration of the Healthy Jeart application was conducted. This involved presenting the application to the teachers at the school where the project was conducted. Additionally, parents of the young participants were provided with a comprehensive overview of the application’s features. Parents were given ample opportunity to seek clarifications and ask questions regarding the application. Once fully informed, their voluntary and written consent was sought for the participation of their children in the study. The informed consent process aimed to ensure that parents were fully aware of the study’s objectives, procedures, potential benefits, and any associated risks. Upon obtaining parental consent, the Healthy Jeart app was then introduced to the young participants. They were encouraged to actively engage with the application by installing it on their personal devices, such as mobile phones or iPads. This participatory approach aimed to foster a sense of ownership over and familiarity with the application, promoting genuine and voluntary involvement. Throughout the study, ongoing ethical considerations were paramount. The privacy and confidentiality of the participants were rigorously maintained, and any concerns or queries raised by participants, or their parents, were promptly addressed.

Thus, the app was first introduced and utilized within the school premises. Subsequently, the young participants were allotted a two-month period to independently use the app either within or outside the school setting. Succeeding this period, the students were re-engaged, and the usability evaluation tool was implemented within the classroom environment. Participants were prompted to complete a questionnaire designed to gather insights into their views on facets concerning the app’s content, usefulness, browsing experience, and feedback mechanisms. The questionnaire encompassed an evaluation of the app’s overall ease of use and clarity and how it contributed to their understanding of healthy behaviors. This stage of the research occurred during the opening semester of 2022.

To meet the inclusion criteria, the young individuals were required to possess an electronic device capable of installing the app, secure informed consent from their parents, willingly participate in the research, and provide responses to more than 90% of the questions. Five students declined participation in the research, while 84 students completed less than 90% of the questionnaire, resulting in a final count of 101 participants.

### 2.2. The Usability Evaluation Tool

Before conducting the usability analysis of the Healthy Jeart mobile application, preparatory steps were taken to identify the characteristics linked to usability. This involved tailoring the metrics typically linked to these attributes, as outlined in ISO 9241-11 [31]. Additionally, a questionnaire was developed specifically for this study’s purposes.

When formulating the questionnaire, the content validity was ensured using a rigorous process. This involved selecting items derived from literature research, the researchers’ expertise, and consultation with field experts, resulting in a pool of 30 items. This set of items underwent scrutiny by a panel of three university professors specializing in education, computer science, and nursing. Their task was to evaluate the items’ quality, eliminating any ambiguities or any deemed inappropriate and determining their alignment within pre-established facets (Content, Navigation, Utility, Feedback, and Overall appraisal).

From this assessment, facets for which there was no consensus among the jury members regarding the distribution of items were indicated. As a result, 8 items were eliminated due to a consensus of more than 50% between the panel members. The study left 22 items (C1–C22) distributed among the following facets: Content (6), Navigation (6), Utility (4), Feedback (4), and Overall Appraisal (2) on a 6-level Likert scale, from strongly disagree to strongly agree, except for question C18, which was dichotomous (Yes/No). The answer to question C19 depends on the answer to the previous question and is only considered for participants who answered “Yes” to question C18. A score of 4 marks the cut-off point, distinguishing satisfaction from dissatisfaction. Note that for questions C6 and C9, due to their wording, the rating scale was inverted.

We chose a six-point Likert scale due to its increased sensitivity compared to five points, its balanced midpoint to reduce bias, and its ideal balance of detail and simplicity, making it easier for respondents to use compared to a seven-point scale.

Throughout this stage, our primary emphasis was on evaluating the user friendliness of the Healthy Jeart app. Simultaneously, we meticulously scrutinized various aspects related to the tool’s accuracy, specifically its capability to precisely measure the targeted concept—in this instance, the health behavior endorsed by the app. As previously mentioned, our tasks encompassed defining the concepts, conducting an exhaustive literature review, and collaborating with experts to ensure the concepts’ representation was adequate. By applying the tool to a subset of young participants, as evidenced by the presented data, we anticipate identifying potential issues and guiding subsequent adjustments to enhance both clarity and precision. Finally, employing Cronbach’s alpha yielded a value of 0.703 for the overall scale (due to their distinct characteristics, the C18 and C19 items were omitted from the reliability analysis). Furthermore, our analysis confirmed that the removal of individual items did not contribute to a further improvement in the alpha value, signifying the stability of the scale.

Table 1 presents the description of the usability questionnaire.

Alongside the 22 questions mentioned earlier, the data collection tool encompassed an initial section aimed at profiling the participants. This section gathered information regarding their age, gender, education, experience level, and typical usage patterns with similar applications.

## 3. Results

### 3.1. Sociodemographic and Electronic Device Usage Characterization

A total of 101 students, representing both primary and secondary school levels, participated in evaluating the Healthy Jeart app. The gender distribution was nearly equal, with 49.5% girls and 50.5% boys. The group’s average age was 13.27 years, ranging from the youngest participant at 11 to the oldest at 17. The majority (80.2%) of participants were enrolled in secondary school.

Regarding electronic device usage, our findings revealed that a significant majority (52.5%) spend between 2 to 4 h daily using screens. Their screen time serves both entertainment and study purposes, although 33.7% of participants exclusively use these devices for entertainment. In Spain, particularly at the school where this study occurred, students are usually not permitted to use mobile phones during regular classes, with exceptions being infrequent and primarily for educational reasons. Consequently, when we refer to time in this context, it pertains to periods outside of regular school hours.

Table 2 displays the sample’s sociodemographic traits along with their patterns of electronic device utilization.

### 3.2. Usability Analysis

We scrutinized the data collected from the questionnaire administered to 101 students, categorizing them across the various facets (Table 3). Our focus was on highlighting the percentage of responses surpassing 4 (the satisfaction threshold) for each item in the tool, along with providing descriptive statistics like mean, standard deviation (SD), mode, minimum, and maximum values. Notably, questions C18 and C19 underwent a distinct analysis due to their unique answer formats.

It is important to highlight that among the 20 examined items, 17 (85%) showcase over 90% of responses above 4, indicating widespread satisfaction within the majority of the sample. Only items C6 and C12 demonstrate comparatively lower satisfaction percentages, with the latter being the sole item where less than 70% of students scored above 4. Therefore, a considerable proportion of young individuals find the information within the app to be overly extensive and difficult to comprehend (C6). Additionally, they encounter challenges in identifying the entity that funded the Healthy Jeart app (C12). Notably, question C21 exhibits 100% of responses surpassing the cut-off value. Questions C2, C5, C13, and C20 were also very close (99%) to unanimity in terms of response.

Analyzing the different facets globally, we see that the average response is higher than 5 in all of them, which would correspond to above the moderately satisfied level, with the feedback facet showing the best values ($\bar{x}$ = 5.56), followed by the Utility ($\bar{x}$ = 5.46), Overall Appraisal ($\bar{x}$ = 5.44) Content ($\bar{x}$ = 5.34), and finally Navigation ($\bar{x}$ = 5.23) facets.

Regarding question C18 within the Feedback facet, we found that 27.7% of respondents asked questions or made some kind of comment directed at the technical team, and of these, only 46.43% received feedback (C19).

## 4. Discussion

Educators and healthcare practitioners working with young individuals need to embrace fresh roles and perspectives to effectively address modern challenges. This includes adapting to the impacts of globalization and the increasing technological advancements shaping our world.

The potential of m-health apps to attract young users is now acknowledged, given their tech-savvy nature and extensive smartphone usage. When creating and launching m-health apps, it is crucial to take into account the requirements and choices of young users. Presently, experts stress the significance of education and awareness campaigns directed toward the younger demographic, aiming to boost the adoption of m-health apps and foster healthy behaviors. [32]

Educational institutions serve as crucial hubs for preventing health issues and fostering healthy lifestyle habits. Both schools and families play active roles in averting the health problems tied to poor dietary choices and a lack of physical activity, habits often ingrained from early years. Within the action plans of educational establishments, the influence of social and digital media on learning processes is an undeniable factor that cannot be overlooked [17].

In this scenario, the principal aim of the Healthy Jeart app is to spread awareness about healthy dietary behaviors and encourage participation in physical activity, all geared toward enhancing the well-being of primary and secondary school students [17].

Once a mobile health application is crafted, evaluating its usability before its public release becomes crucial. The evolution of numerous smartphone apps in response to this advancement has significantly impacted health-related concerns. To incite user motivation for app adoption, prioritizing usability from the outset and continuously assessing it during the development phase is imperative. This approach aims to mitigate potential usability issues upon the applications’ release [33].

In this setting, to address the necessity of assessing the usability of the Healthy Jeart app, we designed a questionnaire comprising 22 questions distributed across five different facets. This questionnaire was introduced to parents and teachers initially and subsequently administered to a cohort of 190 primary and secondary school students from an Andalusian school.

The sample consisted of 101 youths, with an almost equal representation of both genders, averaging around 13 years old, and predominantly attending secondary school.

Upon scrutinizing the electronic device usage patterns, it was observed that most young individuals utilized them for approximately 2 to 4 h daily. Among these, one-third dedicated this time solely to entertainment activities, while nearly 60 percent employed such technology for a combination of school-related tasks and leisure pursuits. These findings are consistent with the data found internationally, as can be seen in the OECD reports in particular [34].

In OECD nations, there has been a consistent rise in the number of children with both Internet access at home and access to various digital devices. Computers were initially the preferred tool for young individuals to use to access the Internet; nevertheless, the trend has shifted gradually, with devices like tablets and smartphones gaining more popularity for online activities than computers. Observing the habits of 15-year-olds in OECD countries, it is evident that they spend roughly two and a half hours online outside of school on an average weekday. However, this duration increases to over three hours on a typical weekend day [34]. Additionally, it is evident that their connectivity is not limited to the home environment alone, as children also make use of mobile technologies while on the move and during school hours [35].The usability assessment questionnaire for the Healthy Jeart app was created by tailoring its content to the goals and intended audience while drawing upon the principles outlined in the ISO 9241-11 standard [31]. The foundation for deriving usability measures, as detailed in ISO 9241-11, involves specifying the intended goals, describing the context of use, and defining the target values of effectiveness, efficiency, and satisfaction. All these elements were previously established for the Healthy Jeart app. In accordance with ISO 9241-11, by taking into consideration these components, organizations can create usability measures customized to specific goals, context of use, and user requirements. This, in turn, streamlines the evaluation and enhancement of product usability in practical work settings. Given that usability pertains to how well a system, product, or service can be utilized by designated users to achieve particular goals effectively, efficiently, and satisfactorily within a defined context, it can be affirmed that the app under evaluation in this study meets the principles outlined in the standard [31]. On a satisfaction scale ranging from 1 to 6, only questions C6, concerning the depth and volume of information within the app, and C12, regarding the ease of locating information about the project’s funding organization, scored below 5. However, they both remained above the satisfaction cut-off value of 4. The overall average satisfaction score across all questions is 5.4, indicating a highly satisfactory level. These elements of usability, essential for user adherence to health apps, have been recognized by other authors as well [36,37,38].Examining the various aspects of and corresponding queries within the tool, it is apparent that the measure of effectiveness, evaluating the precision and comprehensiveness with which users accomplish specific objectives, was satisfactorily met. This is notably evident, for example, in the satisfaction level associated with question C22 (96.1%), indicating that engaging with the Healthy Jeart app enables the exploration of previously unknown aspects of healthy behaviors. This aligns with the World Health Organization’s [39] assertion that m-health solutions have the potential to enhance the health and well-being of teenagers by offering accessible and convenient healthcare information.

Lastly, when assessing efficiency—measured by the relationship between the resources expended and outcomes gained—remarkably favorable values were attained. This is notably apparent, especially in question C5 (99%), which focuses on the clarity of information delivery across the app and its ease of retention.

It should be noted that questions C18 and C19 need to be reworded so that they can be assessed using the same measure as the other parameters. We also think that the inverse wording of items C6 (76.2%) and C9 (86%) may have made it difficult to interpret and answer, so they will also be rewritten.

To enhance the Healthy Jeart app, we will revamp the information structure for a more user-friendly experience. We will add detailed explanations and revise the FAQs and guides, along with links to external resources. The search feature will be improved for easier navigation. A dedicated section will be created for Project Funding Information, offering comprehensive details about the funding organization, including the mission, vision, and contact information, with direct links to its official website for additional information.

We will implement an improved feedback system within the application to facilitate users in expressing their opinions on the provided information and proposing enhancements. Our commitment extends to regular updates, ensuring the currency and relevance of information. Users will be promptly notified of updates and improvements through push notifications or in-app messages. Additionally, we intend to conduct comprehensive usability testing involving a diverse user group from various contexts to identify and rectify any potential issues related to navigation or information accessibility. By addressing these components, we are confident in our ability to significantly enhance the user-friendliness and informativeness of the mobile application, thereby elevating the overall user experience and satisfaction.

## 5. Conclusions

In response to recent events like the pandemic and widespread home confinement, there is a growing interest in developing innovative methods to help teachers and other educators engage students using virtual activities. This includes promoting health initiatives among the younger population by enhancing skills, aligning with the Digital Spain 2025 plan’s [40] commitment to widespread digital transformation. Society’s focus on youth-related issues in schools, crucial influencers of students’ lives, emphasizes health concerns like unhealthy habits leading to diseases. The Healthy Jeart app, with its motivating features, serves as a tool to foster positive health habits in adolescents, benefiting both educators and healthcare practitioners. To delve more profoundly into assessing the application’s quality, this paper highlights the results derived from a comprehensive usability analysis. This examination delves into specific attributes tied to the tool’s content, user experience and utility and the feedback it offers. Using the gathered data, we can precisely identify the application’s strengths and weaknesses.

In terms of strengths, we include the ability to convey which healthy habits are advisable and which are not, the clarity of the content, the accessibility during navigation, the quality of the installation tutorial, and the clear identification of the objectives to be achieved using Healthy Jeart.

Among its possible weaknesses, which therefore show a path for improvement, we highlight that the volume of information provided by the application can be very dense and the mechanisms for confirming receipt of messages and suggestions should be more successful and faster. The first of these aspects will be addressed in a future update. Since this result came to light, the second has already been taken into consideration, as it mainly affects the support team, tasked with responding more quickly, as practically all the suggestions and comments have been answered since the application was rolled out. While our study provides valuable insights into health-related applications, it is essential to recognize and address certain limitations inherent in our methodology. Convenience sampling, which we employed, entails limited control over variables that could potentially influence the study outcomes. This lack of control poses challenges in isolating specific factors and establishing definitive causal relationships.

Another critical consideration pertains to the extrapolation of our findings to other health-related applications. We acknowledge the significance of this concern. While the questionnaire was specifically crafted for a particular application, certain aspects of its structure and questions may be adaptable to more widespread use. However, we advise exercising caution in direct extrapolation, as the questionnaire’s effectiveness could vary based on the specific context, features, and objectives of other health-related applications. Further research and validation would be imperative to assess its applicability beyond the scope of our current study.

## 6. Contributions and Future Directions

This research has significant implications for various stakeholders, including health professionals, teachers, parents, and other educators working with teenagers. Health practitioners can leverage insights from the Healthy Jeart app’s usability analysis to recommend effective digital tools for promoting positive health habits among adolescents. Teachers can use the findings to enhance virtual engagement and integrate digital health applications into educational strategies. Parents and educators gain clarity on app strengths and weaknesses, aiding informed decisions on incorporating digital health tools into daily routines.

The study lays a foundation for future research on digital health applications, offering a framework for evaluating their usability and effectiveness. The commitment to addressing app weaknesses reflects a culture of continuous improvement, setting a precedent for developers to prioritize user feedback.

This cross-disciplinary research fosters collaboration between technology, health, and education, addressing multifaceted challenges in promoting adolescent health. In conclusion, the study provides valuable insights and a practical framework and sets the stage for future research at the intersection of technology, health, and education.

## 7. Patents

Safe Creative intellectual property registration took place on 29 November 2019 at 9:29 UTC, with the registration of the work with the code 1911292583005 and title “Healthy Jeart” made by the user with code 1909023179153.

## Figures and Tables

**Figure 1 healthcare-12-00408-f001:**
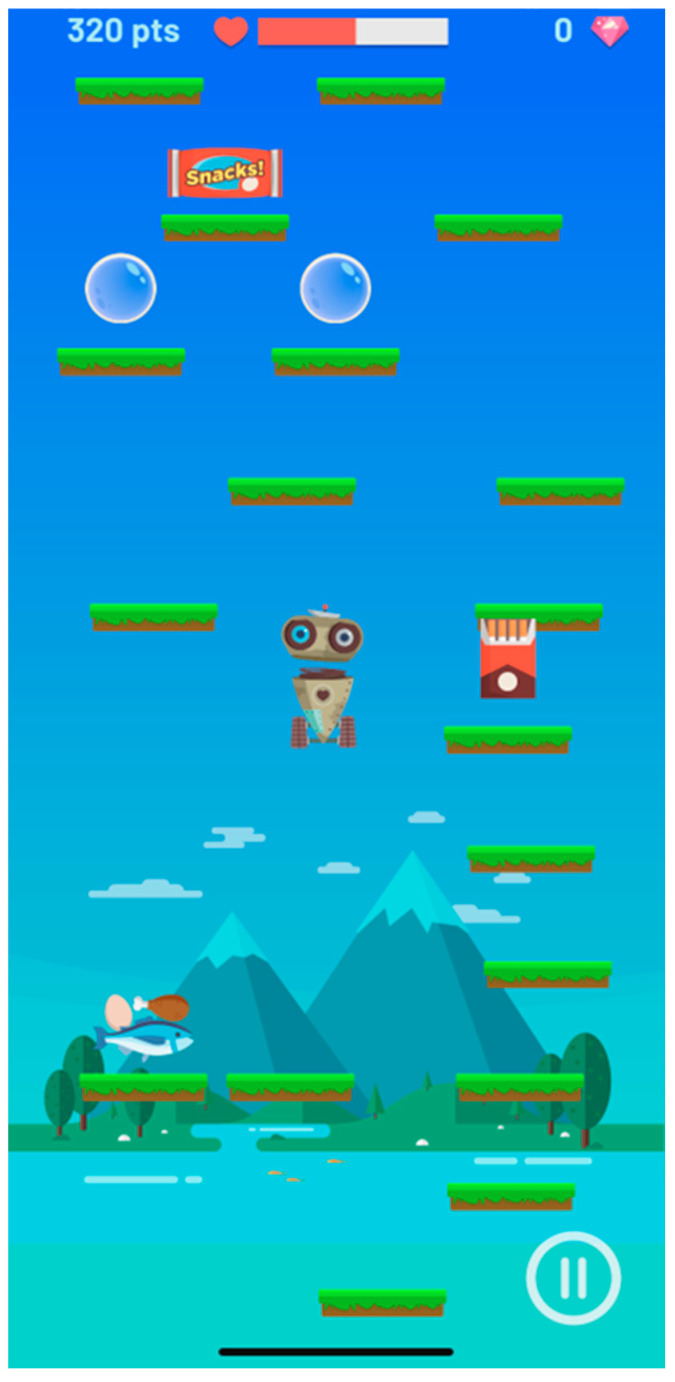
A game to work on healthy habits playfully.

**Figure 2 healthcare-12-00408-f002:**
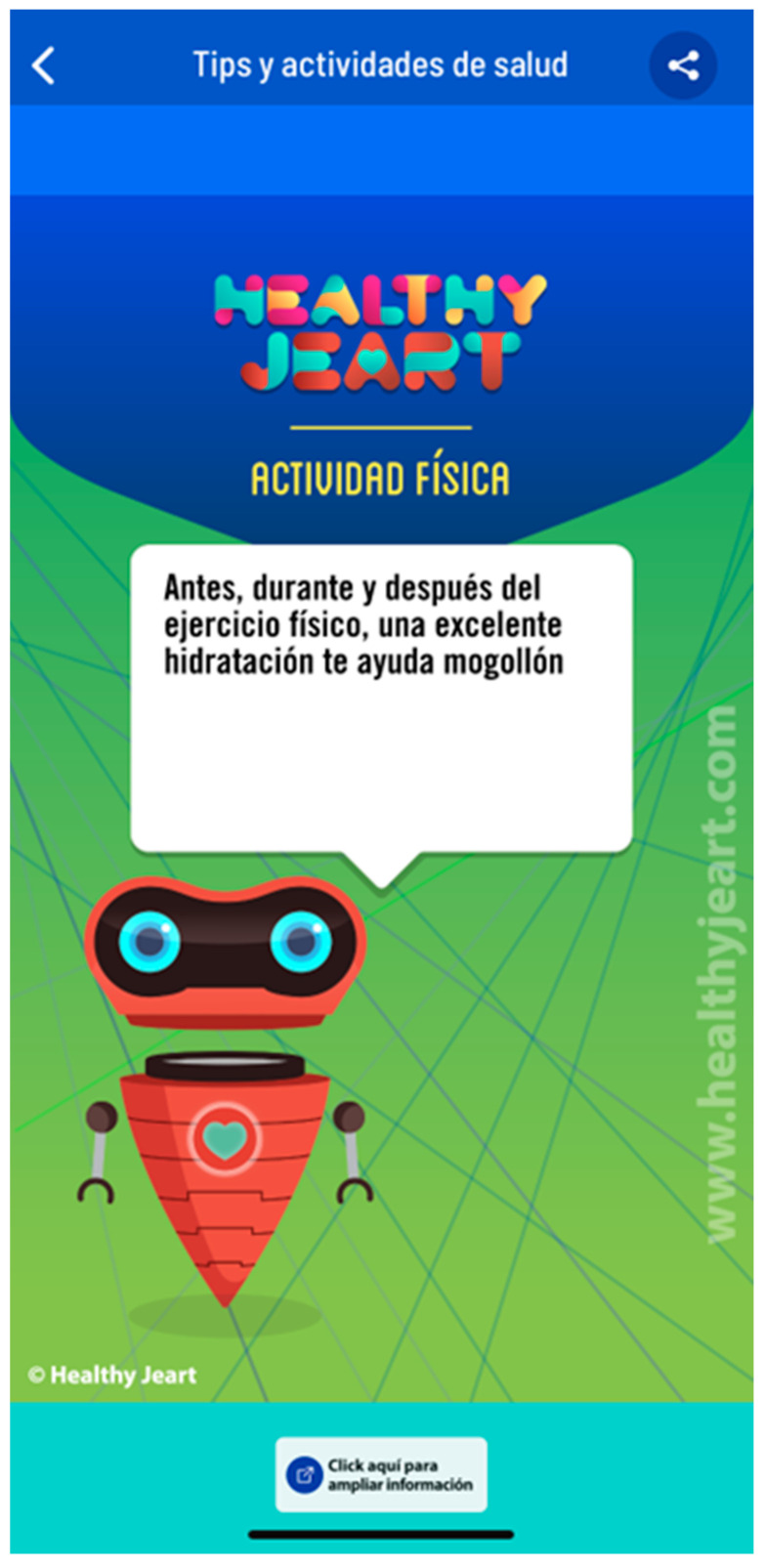
Healthy tips and didactic resources. “Before, during and after physical exercise, excellent hydration helps you a lot”.

**Table 1 healthcare-12-00408-t001:** Usability questionnaire design.

Facet	Question
**Content**	
**C1**	When the Healthy Jeart app opens, I find the options highlighted in the main menu sufficient and I don’t notice anything missing.
**C2**	After using the app, I can now easily distinguish between healthy and unhealthy habits.
**C3**	The distribution of the contents of the app (texts, images, test…) seems good to me.
**C4**	The texts used to access the contents are sufficiently descriptive of what is offered through them.
**C5**	When I access the information in Healthy Jeart, it is presented clearly so that it is easy to understand and remember.
**C6**	The information presented in the app is too extensive and hard to assimilate.
**Navigation**	
**C7**	You can easily and clearly see the options you are browsing through in Healthy Jeart.
**C8**	There are elements that let you go back in a clear and simple manner.
**C9**	You remember seeing some type of advertisement in the app.
**C10**	The operating speed of the app is good.
**C11**	In the app, the tasks of navigating or moving around the app, clicking on buttons, selecting options, etc., are done in the same way throughout the app.
**C12**	I know who funded the Healthy Jeart app.
**Utility**	
**C13**	After first contact, the objectives of Healthy Jeart were clear to me.
**C14**	You can identify what content/information and services the app provides so that you could list some of them.
**C15**	The content/information provided by the app is useful to me.
**C16**	Overall, I was positively surprised by the app.
**Feedback**	
**C17**	It is easy to find out how to make suggestions for improvement or comments to the Healthy programmers.
**C18**	Did you send any suggestions or comments about the application?
**C19**	If so, you received a message confirming that it had been received successfully; how satisfied were you with the answer?
**C20**	The tutorial easily resolves any possible doubts you may have about how to use the Healthy Jeart app.
**Overall appraisal**	
**C21**	My overall rating of the app’s ease of use and clarity is:
**C22**	Interacting with Healthy Jeart enabled me to discover aspects of healthy behaviors that were previously unknown to me.

**Table 2 healthcare-12-00408-t002:** Sociodemographic and electronic device usage description.

Variables	Frequencies n = 101 n (%) Average (SD)
**Gender**	
Female	50 (49.5%)
Male	51 (50.5%)
**Age**	
Range 11–17 years	
$\bar{X}$ (SD)	13.27 (1.321)
$M_{o}$	13
**School Grade**	
Primary	20 (19.8%)
Secondary	81 (80.2%)
**Daily engagement with electronic devices**	
1–2 h	23 (22.8%)
2–4 h	53 (52.5%)
4–8 h	25 (24.8%)
**Screen time or device interaction main goal**	
Entertainment	34 (33.7%)
Studying	9 (8.9%)
Both	58 (57.4%)

**Table 3 healthcare-12-00408-t003:** Healthy Jeart usability questionnaire analysis.

Question	>4 (%)	$\bar{X}$	SD	Min.	Max.	$M_{o}$	$\bar{X}$
**Content**							
C1	91.1%	5.27	1.148	1	6	6	5.34 (0.571)
C2	99%	5.56	0.684	3	6	6	
C3	96%	5.49	0.856	2	6	6	
C4	97.1%	5.46	0.995	1	6	6	
C5	99%	5.62	0.691	2	6	6	
C6	76.2%	4.63	1.848	1	6	6	
**Navigation**							
C7	98.9%	5.67	0.634	3	6	6	5.23 (0.597)
C8	96.1%	5.51	1.064	1	6	6	
C9	86%	5.27	1.643	1	6	6	
C10	96.1%	5.28	0.918	1	6	6	
C11	94.1%	5.35	1.053	1	6	6	
C12	69.3%	4.33	2.103	1	6	6	
**Utility**							
C13	99%	5.68	0.720	1	6	6	5.46 (0.561)
C14	93.1%	5.31	1.120	1	6	6	
C15	97%	5.49	0.890	1	6	6	
C16	95%	5.38	0.958	1	6	6	
**Feedback**							
C17	92.1%	5.38	1.028	1	6	6	5.56 (0.561)
C18 *	No		72.3%	Yes	27.7%		
C19 *	Strongly satisfied			46.43%			
C20	99%	5.74	0.560	3	6	6	
**Overall appraisal**							
C21	100%	5.67	0.550	4	6	6	5.44 (0.760)
C22	96.1%	5.20	1.175	1	6	6	

* Statistical analysis differs based on the question’s nature.

## Data Availability

The data presented in this study are available on request from the corresponding author. The data are not publicly available due to privacy restrictions.

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
