# Peer review of "Assessing the Hands-on Usability of the Healthy Jeart App Specifically Tailored to Young Users"

_healthcare, 2024, doi:10.3390/healthcare12030408_

Round 1

Reviewer 1 Report

Comments and Suggestions for Authors

This paper describes the usability study of a mobile App in the health domain. For this study, the end-user population is teenagers. The paper follows a standard structure, is readable, and seems appropriately referenced. Sufficient information is included for reproductivity. Several issues need further consideration before the paper can be considered for acceptance.

1.      A better justification for the choice methodology is needed. The approach adopted is selected for its applicability in mobile learning apps. Yet it is not clear what “learning” was evaluated. The usability methodology, as described, seems relatively standard.

2.      Typo in line 50.

3.      While the methodology is apparently aligned with ISO9241-11, this alignment is unclear in the methodology and only hinted at in the discussion.

4.      Why a six-level Likert scale rather than five or seven?

5.      It is difficult to visualize the Healthy Jeart App. Some additional information and screen snapshots would be helpful.

6.      While results are adequate, it is difficult to see how they inform practice and further research in this domain. 

Comments on the Quality of English Language

English seems fine. 

Reviewer 2 Report

Comments and Suggestions for Authors

Thank you very much for the opportunity to review this article. The text is engaging as it addresses a relevant and understudied topic in the young population and focuses on the use of new technologies, something fundamental for today's young generations. However, I think several minor aspects could be revised to improve the overall quality of the text:

The abstract can be improved by structuring it more systematically, separating the different sections better, and offering more details—for example, some details of the methodology and a summary of the main results.

Considering that the length of the article is short, which is a positive thing, I am struck by the length of the introduction, which is somewhat disproportionate to the rest of the text. If possible, it would help a little if the authors managed to slightly reduce its length to allow readers to enter more fluently into the rest of the text since, I insist, brevity is one of its strong points.

Given that this is a survey-based study, in terms of methods, it would be helpful for the authors to provide a more detailed overview of the main characteristics of the eligible population. This is essential to assess the inevitable selection bias inherent in this study. Another crucial aspect is better describing the sampling method employed, as it also contributes to this potential selection bias.

Another important aspect related to the methodology is that it is better to define some of the questionnaire's fundamental elements. For example, looking at the design of the questionnaire and the questions it contains, the questionnaire assesses qualitative aspects, i.e., subjectively perceived by the user. In that case, a validation process could have been carried out using exploratory and confirmatory factor analysis, which help to understand, in a given context, the test's validity, which is the tool used for this study. Therefore, it would be helpful for the readers if the authors could better define the questionnaire's process and whether validation studies were conducted.

It would also be helpful for the authors to detail the ethical aspects better. For example, an ethics committee may have assessed the study since a survey involves interaction with study participants.

The authors should add a specific section on limitations in the discussion. Although they cite some of them indirectly, mentioning the limitations inherent to this type of study and others related to their particular work is helpful. In particular, I encourage them to discuss potential selection bias, as this is one of the main limitations of the study, as well as unavoidable. They should be able to assess the external validity of their findings in terms of this and other limitations.

The conclusions also seem lengthy compared to the rest of the text, so I recommend summarizing them.

Reviewer 3 Report

Comments and Suggestions for Authors

I am grateful for the opportunity to review this manuscript on a digital solution for young users. Please find my suggestions and questions.

MANUSCRIPT

-Please, enter the date of study in the Abstract.

-Please, enter the date of the different steps of the study in the Methods section:

Lines 119-125: “Initially, the app was introduced and utilized within the school premises. Subsequently, the young participants were allotted a two-month period to independently use the app either within or outside the school setting. Succeeding this period, the students were re-engaged, and the usability evaluation tool was implemented within the classroom environment. Participants were prompted to complete a questionnaire designed to gather insights into their views on particular facets concerning the app's content, usefulness, browsing experience, and feedback mechanisms. “

-The objectives could be reviewed:

Line 17: “This work constitutes a preliminary analysis of the usability perceived by users of the application.”

Lines 101-105: “Hence, this project focuses on refining the usability of the Healthy Heart application. Its primary aim is to pinpoint any usability concerns, grasp user requirements, and ultimately enhance the overall userfriendliness of the product. Evaluation of the usability of Healthy Jeart was carried out empirically, with real users (school-age adolescents)”

Lines 298-299: “To explore deeper into evaluating the application's quality, this paper showcases the outcomes of a usability analysis”

INTRODUCTION

Lines 66-99: This text is relevant but does not link to the previous paragraphs about the Health Jeart app and healthy behaviors. The authors could subdivide this section into healthy behaviors among young people and another item on digital applications and the importance of their usability.

MATERIAL AND METHODS / RESULTS

Line 109. “This is an exploratory and descriptive study…”  Exploratory study is used for qualitative studies. The authors applied a questionnaire (quantitative study), with an open response (C18) and did not present an analysis of its content (Lines 206-208).

-Lines 110-112: “The population consisted of  190 SECONDARY school students from a public-private school in the province of Andalusia  (Spain), who installed the Healthy Jeart app on their electronic devices” However, the authors described primary and secondary school participants in Table 2.

-Did the questionnaire provide neutral answers (I do not know/ I do not answer)?

- Attention: With the exception of questions C6, C12, C18 and C19, the questions presented positive content (in favor of the way the application is available) which may influence participants to agree with the question. Neutral questions should have been used (for example, “To what extent do you think the main menu is sufficient and anything is missing?”)

-Table 2: “Daily engagement with electronic devices:  1- 2 hours”   Is this information on outside school hours? Please describe in the text.

DISCUSSION

The discussion section needs to be rewritten:

·         The first paragraph of the discussion should provide information about the study and most important results.

·         The current discussion looks like a summary of the results.

·         The authors could discuss their results with those from the literature.

·         The authors also did not include the limitations of the study.

CONCLUSION

The conclusion section needs to be rewritten:

·         Lines 282-297: This information could be described in Introduction section.

·         Line 198:  “To explore deeper into …”     To explore deeper is for qualitative study.

·         The strengths and weaknesses could be described in Discussion section.

Reviewer 4 Report

Comments and Suggestions for Authors

Dear authors. I greatly appreciate the opportunity to have been able to review your work. Promoting healthy lifestyles in young people is essential for their maintenance in the future and doing so through apps facilitates the achievement of objectives since today's young people make substantial use of digital media.

Next I would like to make a series of comments/suggestions.

- The introduction seems adequate and complete to me. It perfectly addresses the topic of study. Its structure and organization make it easy to read and understand the topic.

- Has the Healthy Jett application been created by health professionals? Who is involved in its design and development? I have seen that it is endorsed by several agencies and associations, which increases the value of said application. It's a simple curiosity.

-Why have you decided to study this application and not others with a similar theme? Has it been created or developed by you?

- Regarding the quality and usability of mobile applications, I recommend reviewing the MARS (quality) and uMARS (usability) applications, both tools validated in the Spanish language.

- Methodology. Did all students have mobile devices or did some students download the app on their parents/guardians' phones?

I would like to be able to read what application usability tool they have used, what theoretical framework they have based it on or what questions they asked the students to obtain the usability data.

What do you think is the reason why some students have not completed the survey or others have not even wanted to take it?

Why have you decided to carry out your own questionnaire when there are validated tools to evaluate the usability of mobile applications? Scientific evidence already recommends some and there is experience in their use.

Have you considered the validation of the questionnaire? Can the use of said questionnaire be extrapolated to other health-related applications? Have these questions been raised?

Where does, from your point of view, lie the low adherence of children and young people to healthy behaviors related mainly to diet and physical activity? I believe that this research opens the door to debate and constructive criticism with the results they offer. Do you think that the use of applications will only help teachers? In the conclusion they indicate this and I believe that this statement can be improved.

I appreciate that you included the limitations of the study.

I suggest you also include a section that indicates what this research contributes to the scientific community and future lines of research.

I appreciate your work, I found it very interesting. Thank you.

Round 2

Reviewer 1 Report

Comments and Suggestions for Authors

1. Line 58. Combine the first two lines of this paragraph.

2. There are several instances throughout where two full stops appear in succession in Line 87. Full stop missing in line 330. Space missing in 390.

3. Add a line justifying the choice of 6 point Likert scale.

Comments on the Quality of English Language

English is fine. 

Author Response

We greatly appreciate your effort in reviewing our manuscript. In response to your points, we have implemented the recommended modification.